# Detection of High-Performance Wheat Genotypes and Genetic Stability to Determine Complex Interplay between Genotypes and Environments

Ibrahim Al-Ashkar [1,2,*], Mohammed Sallam [1], Khalid F. Almutairi [1], Mohamed Shady [3], Abdullah Ibrahim [1] and Salem S. Alghamdi [1]

1   Department of Plant Production, College of Food and Agriculture Sciences, King Saud University, Riyadh 11451, Saudi Arabia
2   Agronomy Department, Faculty of Agriculture, Al-Azhar University, Cairo 11651, Egypt
3   Department of Agricultural Engineering, College of Food and Agriculture Sciences, King Saud University, Riyadh 11451, Saudi Arabia
*   Correspondence: ialashkar@ksu.edu.sa

**Abstract:** Abiotic stress decreases crop production worldwide. In order to recommend suitable genotypes for cultivation under water deficit and heat stress conditions, an overall understanding of the genetic basis and plant responses to these stresses and their interactions with the environment is required. To achieve these goals, the multitrait genotype-ideotype distance index (MGIDI) was utilized to recognize abiotic-stress-tolerant wheat genotypes, and the weighted average of absolute scores (WAASB) index as well as the superiority index, which enables weighting between the mean performance and stability (WAASBY), were utilized to recognize high-yielding and stable genotypes. Twenty wheat genotypes were examined to determine the abiotic stress tolerance capacity of the investigated genotypes under nine test environments (three seasons × three treatments). Abiotic stress significantly decreased most morpho-physiological and all agronomic traits; however, some abiotic-stress-tolerant genotypes expressed a slight reduction in the measured traits as compared with the control group. G04, G12, G13, and G17 were identified as convenient and stable genotypes using the MGIDI index under all environments. Based on the scores of the genotype index (WAASB), G01, G05, G12, and G17 were selected as superior genotypes with considerable stability in terms of the grain yield (GY). G04, G06, G12, and G18 were classified as cluster (I), the productive and stable genotypes, using the WAASBY superiority index. The combined indices (MGIDI and WAASB) and (MGIDI and WAASBY) revealed genotypes G12 and G17 and genotypes G04 and G12, respectively, as the most stable candidates. Therefore, these are considered novel genetic resources for improving productivity and stabilizing GY in wheat programs under optimal conditions, water deficit, and heat stress. The genotype G12 was jointly expressed in all three indices. Stability measures using WAASB may help breeders with decision-making when selecting genotypes and conducting multi-environment trials. Hence, these methods, if jointly conducted, can serve as a powerful tool to assist breeders in multi-environment trials.

**Keywords:** abiotic stress; bread wheat; high-performance; genetic stability; MGIDI index and Multi-environment trials (METs)





## 1. Introduction

Agricultural productivity decreases due to many environmental factors, including desertification, degradation, salinization, land pollution, and a lack of water resources, threatening sustainable crop production. The risk may surge when multiple stresses coincide, for example, a raised temperature and water shortage [1–3]. For these reasons, with the continuous increase in the global population, an increase in the production of major cereals (wheat, rice, and maize) by 2–3% each year is essential to achieve food stability and

security. Temperature increase and a lack of water resources adversely influence physiological processes, which is directly reflect in the yield. A raised temperature remarkably reduces the wheat yield by as much as 35% when it is 3–4 °C above normal throughout the grain filling period [2–5]. Moreover, a water shortage reduces growth and yield traits [2,6]. Therefore, breeders and geneticists in wheat breeding programs are required to collaboratively increase their productivity and boost wheat tolerance to multiple stresses through the development of new genotypes with enhanced characteristics, including a high-yielding performance and tolerance of multiple stresses [3]. Potential genotypes must be, therefore, evaluated in terms of their agro-physiological traits and genetic parameters, alongside the use of effective and reliable detection methods for the tolerance of multiple stresses. These detection approaches need to be cost-effective, easy to measure, and quick. In particular, inherited heat and drought tolerance-complicated traits should be a focus [2,3,7,8].

Breeders often take into account a series of morpho-physiological traits that, if combined into a single genotype, could result in higher performance levels (referred as ideotype). The concept behind the ideotype design is improving and increasing the productivity and performance of crops with emphasis on the selection of genotypes bearing multiple traits simultaneously [9–12]. Breeders use linear selection indexes to select high-performance and superior genotypes [13]. The linear phenotypic selection index [14–17] is the Smith–Hazel (SH) index [18,19]. The SH index requires the use of the phenotypic and genotypic (co)variance matrices with an economic weights vector to establish means of coefficients vector indexing for the optimization of the linkages between phenotypic values and unknown genetic values. Due to the use of multiple traits, multicolinearity will certainly exist, leading to poorly conditioned matrices and prejudiced index coefficients, affecting the evaluation of genetic gains. Moreover, breeders often face limitations associated with the economic value determination of the studied traits and the conversion to workable economic weightings [10,15]. Thus, the SH index is not solely sufficient for use in plant breeding trials without the use of other indexes [14,17,20–23]. Consequently, the combination of multivariate techniques is useful for addressing the multicollinearity in multitrait indexes [10,24–26], as it offers an index that covers the weaknesses of the SH index and can be efficiently applied to choose all favorable traits and choose satisfactory gains to be implemented in breeding and biological experiments [27]. Subsequently, Olivoto and Nardino [10] proposed a novel multitrait genotype–ideotype distance index (MGIDI) that focuses on genotype selection and recommendations based on information about multiple traits. The performance of the MGIDI was assessed via a Monte Carlo simulation where the powerful traits with desired gains were computed for multiple scenarios involving different numbers of genotypes and traits, taking into account the nature of the correlations between traits. The MGIDI index requires a two-way table as input data and enables a row ranking according to the desired results in the columns. It has the potential to evaluate more than one dependent trait [28–30] and to assess the strengths and weakness of the selected genotypes [10].

Even though breeders evaluate a set of morpho-physiological traits in their programs, grain yield performance is always considered as the principal trait. Since plants respond to various environmental factors (biotic and/or abiotic), the genotype can perform comparatively well in one environment but poorly in another. This performance may differ in rank across environments, indicating an interaction (GEI) of the qualitative or crossover-type, meaning that special strategies are required for crop improvement. The performance constancy (i.e., the genotype performance shows no rank difference across various environments) denotes whether the quantitative or noncrossover type is present [3,23,31]. To understand the GEI, one must search for new ways of using it and apply it to the selection of high-throughput genotypes, either in specific environments (variety for each region) or under a broad spectrum of environments (variety for all regions). The GEI impact is very important for agricultural researchers and breeders. This impact can adversely influence the correlation between the genotypic and phenotypic value which, in turn, limits the selection of preferred genotypes across environments [3,32]. Therefore, it is crucial to use

stability and adaptability analyses to select preferred genotypes for various environments. Using genotypes in multienvironment trials assists breeders in defining the ideotypes with significant adaptability to various environments [23].

Achieving genuine outputs for the genetic stability and discriminating and interpreting the GEI effect in multienvironment trials for the studied genotypes requires valid and reliable statistical analyses, such as the multivariate analysis, which uses a combination of multiple characteristics with influential computer modeling [8,23,33]. Thus, plant breeders must test the relationships among traits influencing the yield to define accurate selection criteria for use in breeding programs. Older breeders usually try to find models for interactions, known as stability analyses, such as the joint regression [34]. This method was frequently applied by Finlay and Wilkinson [35] and Eberhart and Russell [36]. Later, new methods combining different statistical stability analysis approaches, such as factor analysis (FA) and one-way analysis of variance (ANOVA), termed FANOVA [37], were proposed. This method is currently known as the additive main effect and multiplicative interaction (AMMI) [38]. AMMI analysis is one of the best models to use for the selection of preferred genotypes in most crops [39–44]. Although AMMI analysis offers many advantages for interpreting GEI, the main limitation was noted when analyzing the structure of the linear mixed-effect model (LMM). Hence, a novel model, referred as the weighted average of absolute scores (WAASB), was proposed by Olivoto et al., [23]. The WAASB results from the singular value decomposition of the BLUP (best linear unbiased prediction) matrix for GEI effects generated by an LMM to describe greater ideal genotypes based on a combination of stability and yield performance [23].

The WAASB model combines the characteristic features of the AMMI and BLUP models into one unique index, allowing the selection of stable and high-yielding genotypes of soya bean, barley, maize, and wheat [26,43–46]. The BLUP provides the ability to improve the predictive accuracy and provide reliable estimates of random effects [23,24,47]. Operationally, BLUP and AMMI seem to be distinct approaches that achieve the same goal—to discriminate the GEI pattern from the random error—but they are statistically different. The AMMI analysis keeps a great deal of the GEI pattern in the first interaction principal component axis (IPCA) caused by singular value decomposition (SVD) of the nonadditive effects matrix while keeping a great deal of the random error in the IPCAs of the latter. On the other hand, the BLUP primarily assesses the impacts of the ANOVA model and thereafter returns weights to these influences and, therefore, can be regarded as a contraction estimator [48,49].

Multivariate analysis approaches (multicollinearity, factor analysis, MGIDI index, AMMI, WAASB, and biplots) are frequently utilized for the effective and reliable detection of variation between genotypes, their interactions with environments, and the genotype stability in various environments [2,8,10,23,24,50]. Limited studies have evaluated the impacts of multiple stressors, such as drought and heat, on the mean performance and stability levels of genotypes using the MGIDI index and the WAASB model. Therefore, the principal goals of the current study were to (i) assess 20 wheat genotypes under three (control, drought, and heat) conditions during three consecutive years (the combination of year and conditions generated 9 environments) and the effects of the GEI; (ii) identify the stability traits associated with the mean performance; and (iii) identify genotype (s) with the best performance according to the MGIDI index and the highest stability according to the WAASB model when grown under the three (control, drought, and heat) conditions.

## 2. Materials and Methods

### 2.1. Description of the trial

Six varieties and 14 doubled haploid lines (DHLs) of wheat were cultivated in nine environments (E), as shown in Table 1. The varieties were obtained from the Agricultural Research Center, Egypt, and the DHLs were obtained from the Agronomy Department, Faculty of Agriculture, Al-Azhar University, Nasr City, Cairo, Egypt, as well as from previously published data [51] as described in Table S1 and coded in Table S2. The experiments

were designed in randomized complete blocks (Steel and Torrie, 2000) with three replicates per environment. The plot area, seedling rate, and type of texture soil were described in detail in our previous study [3]. The fertilization rate was 31 kg ha$^{-1}$ P$_2$O$_5$ with the processing of land for cultivation and 180 kg ha$^{-1}$ N with irrigation in batches prior to spike heading. The weather conditions are presented in Table S3.

**Table 1.** Codes used and description of the test for production environments.

| Environment Code | Treatments | Planting Dates | Season |
| --- | --- | --- | --- |
| E1 | Control [full irrigation (100% field capacity), timely sown] | 15 November | 2018/19 |
| E2 | Drought stress [limited irrigation (33% field capacity), timely sown] | 15 November | 2018/19 |
| E3 | Heat stress [full irrigation (100% field capacity), late sown] | 20 December | 2018/19 |
| E4 | Control [full irrigation (100% field capacity), timely sown] | 17 November | 2019/20 |
| E5 | Drought stress [limited irrigation (33% field capacity), timely sown] | 17 November | 2019/20 |
| E6 | Heat stress [full irrigation (100% field capacity), late sown] | 25 December | 2019/20 |
| E7 | Control [full irrigation (100% field capacity), timely sown] | 17 November | 2020/21 |
| E8 | Drought stress [limited irrigation (33% field capacity), timely sown] | 17 November | 2020/21 |
| E9 | Heat stress [full irrigation (100% field capacity), late sown] | 25 December | 2020/21 |

### 2.2. Measurements of Traits

Twenty traits were measured—twelve morpho-physiological traits (photosynthesis rate (Pn), stomatal conductance (Gs) transpiration rate (E), canopy temperature (CT), relative water content (RWC), leaf water content (LWC), flag leaf area (FLA), green leaves area (GLA), leaf area index (LAI), catalase (CAT), polyphenol oxidase (PPO) and peroxidase (POD)), and eight agronomic traits (days to heading (DH), days to maturity (DM), grain filling duration (GFD), number of spikes (NS), plant height (PH) thousand kernel weight (HKW), number of kernels (NKS), and grain yield (GY)), as described in detail in our previous studies [3,52].

### 2.3. Statistical Analyses

#### 2.3.1. Analysis of Variance

The normality of the data was checked to make sure that there were no outliers and that the data followed a normal distribution according to the Shapiro–Wilk test [53]. The three seasons within the same environment type were shown to be homogeneous by Bartlett's test [54]. The data on the 20 studied traits under three environments (seasons × genotypes) were analyzed for each treatment group separately.

#### 2.3.2. Linear Mixed Model

The linear mixed model used to calculate the variance components of the studied traits was

$$Y_{ijk} = \mu + G_i + E_j + Rk_{(j)} + GE_{ij} + \alpha_{ijk} \tag{1}$$

where Y$_{ijk}$ denotes the genotype phenotypic value $_i$ for the trait under study in the environment $_j$ and block $_k$, $\mu$ is the overall mean; G$_i$ is the impact of the $_i$<sup>th</sup> genotype ($i$ = 1, 2, ... , 20); E$_j$ is the influence of the $_j$<sup>th</sup> environment ($j$= 1, 2, ... , 9); R$_{k(j)}$ refers to the impact of the $_k$<sup>th</sup> replication ($k$ = 1, 2, 3); GE$_{ij}$ denotes interaction impact of $i$<sup>th</sup> genotype with the $j$<sup>th</sup> environment; and $\alpha_{ijk}$ refers to refers to the residual error, which was assumed to be normally and independently distributed with a mean of 0 and a variance of $\sigma^2$ [23]. The heritability (h$^2$, broad sense) was calculated from this model as

$$h^2 = \left(\sigma_g^2\right) / \left(\sigma_g^2 + \frac{\sigma_{g \times e}^2}{e} + \frac{\sigma_{re}^2}{r \times e}\right) \tag{2}$$

where g, g × e, $\sigma_{re}^2$, r and e were the genotypic variance; the genotype × environment variance; the residual variance (error); the number of replicates and the number of environments, respectively.

### 2.3.3. Genetic (rg) and Phenotypic (rp) Correlations

The (rp) and (rg) were calculated from Equation (1) as the (co)variance per two traits. The following formulas were used to calculate the (rp) and (rg):

$$rp = cov\ \sigma_p^2 / \sqrt{(\sigma_{px}^2 \times \sigma_{py}^2)} \tag{3}$$

$$rg = cov\ \sigma_g^2 / \sqrt{(\sigma_{gx}^2 \times \sigma_{gy}^2)} \tag{4}$$

where cov $\sigma_p^2$ and cov $\sigma_g^2$ are the phenotypic and genetic (co)variance, respectively and $\sigma_{px}^2$ and $\sigma_{py}^2$ are the phenotypic variances and $\sigma_{gx}^2$ and $\sigma_{gy}^2$ are the genetic variances of trait x and trait y, respectively.

### 2.3.4. MSTI Analyses

The MGIDI was used to rank the genotypes on the basis of multiple trait values, as suggested by Olivoto, et al. [55]. For the first stage, each trait (rX$ij$) was rescaled as

$$rXij = \frac{\eta_{nj} - \varphi_{nj}}{\eta_{oj} - \varphi_{oj}} \times \left(\theta_{ij} - \eta_{oj}\right) + \eta_{nj} \tag{5}$$

where the symbols indicate the following for trait j and genotype *i*: rX$ij$ is the rescaled two-way table; $\eta_{nj}$ is the new maximum value after rescaling; $\varphi_{nj}$ is the new minimum value after rescaling; $\eta_{oj}$ is the original maximum value; $\varphi_{oj}$ is the original maximum value; and $\theta_{ij}$ is the original value for the *i*th genotype. Each column ranged from 0 to 100, considered the desired sense of selection (increase or decrease), and sustained the correlation structure of the original set of variables. The values obtained after rescaling for $\eta_{nj}$ and $\varphi_{nj}$ in a state of positive gains ($\varphi_{nj}= 0$ and $\eta_{nj}=100$) and in a state of positive gains ($\varphi_{nj}= 100$ and $\eta_{nj}= 0$) were used [55]. The second stage was to compute an exploratory factor analysis (FA) through rX$ij$ to group correlated traits into factors and then estimate the factorial scores for each row/genotype/treatment. The scores were then obtained from the data collected for the dimensionality reduction of traits and relationship structure using the following model:

$$F = Z(A^T R^{-1})^T \tag{6}$$

where the letters indicate the following: F is the g × f matrix with the factorial score; Z is the g × p matrix with the rescaled means; A is the p × f matrix of canonical loading; R is the p × p correlation matrix between the traits; g is the number of genotypes; f is the factors retained; and p is the measured traits. The third stage was to compute an ideal genotype. For this, a [1 × p] vector was considered to be the ideotype matrix using the Euclidean distance between the scores of the genotypes, and the ideal genotypes were determined by the MGIDI index, as shown

$$MGIDI = \sum_{j=1}^{f} [\gamma_{ij} - \gamma_j]^{0.5} \tag{7}$$

where the symbols and letters indicate the following: $\gamma ij$ (the score of the *i*th genotype in the *j*th factor ($i = 1, 2, \ldots, t; j = 1, 2, \ldots, f$), where t is the number of genotypes; f is the factors; and $\gamma j$ is the *j*th score of the ideal genotype. The lowest value of the MGIDI indicates that the genotype is more ideal (for all of the measured traits). The selection of all traits was implemented (with a selection intensity of ~20%).

In the METs, the GY data of the wheat genotypes were subjected to WAASB method, which combines features of the AMMI and BLUP methods and is computed as proposed by Olivoto et al. [24]. The WAASB calculation used was

$$\text{WAASB} = \sum_{n=1}^{p} |\text{IPCA}_{ng} \times \text{EP}_n| / \sum_{n=1}^{p} \text{EP}_n \tag{8}$$

where the symbols indicate the following: IPCA*gn* is the score of genotype *g* in the *n*th interaction principal component axis (IPCA) and EP*n* is the amount of variance explained by the nth IPCA. The lowest value of WAASB indicates that the genotype is more stable. The WAASBY index is a superiority index that allows weighting between genotypes' performance (GY) and stability (WAASB index) values, as described by the proposed method presented in [24].

### 2.4. Statistical Software

All statistical analyses were performed using packages in RStudio, R version 4.2.2 (R Core Team 2022). The packages used in this study were "metan", which was used for the multienvironment trial analysis, as per Olivoto and Lúcio (2020), and "cowplot" for used for the graphics arrangement.

## 3. Results

### 3.1. Variability in Genotypes and Traits

The 20 genotypes responded differently to abiotic stresses (drought and heat) compared to normal conditions (control), and considerable variation in the three seasons was seen in the control and abiotic stress treatment groups for almost all traits (Table 2). The genotypes showed significant variation in all traits in the control and abiotic stress treatment groups. The seasons were associated with highly significant variation in most traits, while seven traits (FLA, LWC, RWC, POD, PPO, and CAT) varied among the three treatment groups. Interaction effects (genotypes x seasons) were found to be significant for eight traits in the three treatment groups, nonsignificant for five traits (DH, Pn, POD, PPO, and CAT), and varied for seven traits (DM, GFD, LWC, RWC, NG, Gs, and E) across the three treatment groups (Table 2). Figure 1 displays the plotting performance of the genotypes across the three environments (control, drought, and heat) for 20 traits measured in 20 wheat genotypes as the overall means of the three seasons. All measured traits significantly declined due to abiotic stresses compared to in the control group, except for CT, which increased due to being affected by stress. The performance of the genotypes showed highly significant variation under the same conditions and varied according to the stress type. Most traits showed clear differences between the three treatment groups (control, drought, and heat), while some traits (FLA, Gs, E, POD, PPO, and CAT) showed no clear differences between the abiotic stresses of drought and heat.

**Table 2.** Analysis of the variance of the seasons (S), genotypes (G), and their interactions (S × G) for the 20 studied traits of 20 wheat genotypes measured in three seasons.

| Trait | Unit | Control | | | Drought | | | Heat | | |
|---|---|---|---|---|---|---|---|---|---|---|
| | | **S** | **G** | **S × G** | **S** | **G** | **S × G** | **S** | **G** | **S × G** |
| DH | days | * | *** | ns | * | *** | ns | * | *** | ns |
| MD | days | * | *** | ns | * | *** | ns | *** | *** | *** |
| GFD | days | * | *** | ns | * | *** | ns | *** | *** | *** |
| NS | m$^{-2}$ | *** | *** | *** | *** | *** | *** | *** | *** | *** |
| PH | cm | *** | *** | *** | *** | *** | *** | *** | *** | * |
| FLA | cm$^2$ | ** | *** | * | ns | *** | *** | *** | *** | *** |
| GLA | cm$^2$ | ** | *** | *** | *** | *** | *** | *** | *** | * |
| LAI | | *** | *** | *** | *** | *** | *** | *** | *** | *** |

**Table 2.** *Cont.*

| Trait | Unit | Control | | | Drought | | | Heat | | |
|---|---|---|---|---|---|---|---|---|---|---|
| | | **S** | **G** | **S × G** | **S** | **G** | **S × G** | **S** | **G** | **S × G** |
| LWC | % | *** | ** | *** | *** | ** | *** | ns | *** | ns |
| RWC | % | ns | *** | *** | ** | *** | *** | ** | *** | ns |
| NG | no./spike | *** | *** | *** | *** | *** | *** | ** | *** | ns |
| TKW | gm | *** | *** | *** | *** | *** | *** | *** | *** | ** |
| CT | °C | *** | *** | *** | ** | *** | *** | *** | *** | *** |
| Pn | $\mu mol\ Co_2\ m^{-2}\ s^{-1}$ | *** | *** | ns | *** | *** | ns | ** | *** | ns |
| Gs | $mol\ H_2O\ m^{-2}\ s^{-1}$ | *** | *** | ns | *** | *** | *** | * | *** | *** |
| E | $mmol\ H_2O\ m^{-2}\ s^{-1}$ | *** | *** | ns | *** | *** | *** | *** | *** | *** |
| POD | $U\ g^{-1}\ FW\ mL^{-1}$ | ns | *** | ns | * | *** | ns | * | *** | ns |
| PPO | $U\ g^{-1}\ FW\ mL^{-1}$ | ns | *** | ns | * | *** | ns | * | *** | ns |
| CAT | $U\ g^{-1}\ FW\ mL^{-1}$ | ns | *** | ns | * | *** | ns | * | *** | ns |
| GY | $ton\ ha^{-1}$ | *** | *** | *** | *** | *** | *** | *** | *** | *** |

*, **, and *** indicate significance levels at $p < 0.05$, $p < 0.01$, $p < 0.001$, respectively. ns indicates insignificance. Abbreviations as described in materials and methods

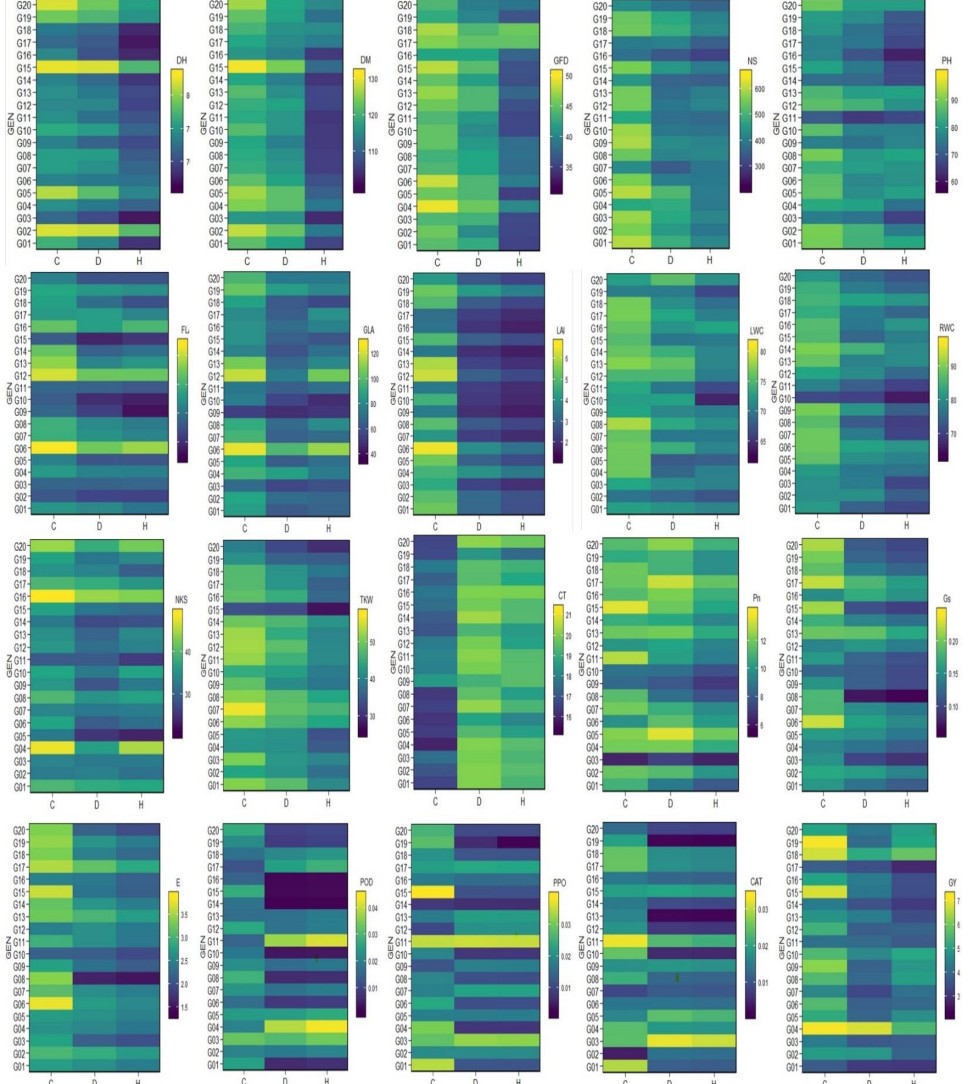

**Figure 1.** Plotting the mean performance of the 20 measured traits as the overall mean across three seasons of 20 wheat genotypes with the three treatment groups: control (C), drought stress (D), and heat stress (H).Abbreviations as described in materials and methods.

### 3.2. Variance Components of Studied Traits

The LR test results exhibited highly significant effects ($p < 0.001$) for all studied traits for both genotype and interaction effects, except for the DH trait for the genotype effect (Table S4). The variance components obtained with the REML were divided into percentages of their contributions to the phenotypic variation in the studied traits, great variation between them was shown. The genotypic variance varied from 17.30% for the LWC to 92.40% for the DH. The G × E interaction varied from 0.07% for the LWC to 47.40% for the CT, and the residual variance varied from 0.40% for the POD to 52.50% for the LWC (Table S4 and Figure S1). For the broad-sense heritability (H2), the computations showed mixed heritability values, which varied from 17.30% for the LWC to 92.40% for the DH. In contrast, the genotypic accuracy of selection (As) for all traits, which measures the correlation between expected and observed values, was high (>87.50%). The genotypic CVs recovered to be higher than the residual CVs for all traits. Consequently, the CVs (g/r) ratio were greater than 1, except for that of the LWC trait. The genotype–environment correlation (rge) showed high values (>0.580) for fourteen measured traits. The high proportion of the genotypic effect in the overwhelming majority of traits indicates that the genotypic effect plays a significant role in the inheritance of the studied traits.

### 3.3. Genetic and Phenotypic Correlations between Studied Traits

To identify the traits related to the GY, the genetic (rg) and phenotypic (rp) correlations between the nineteen traits were estimated with the GY (Table S5 and Figure 2). The results revealed significant positive correlations with nine traits (in descending order, DM, LAI, NS, GFD, DH, Pn, NKS, POD, and PH), negative correlations with seven traits ( in ascending order, E, FLA, CT, Gs, LWC, PPO, and TKW), and three traits were insignificant (GLA, RWC, CAT). The genetic correlations were close to the phenotypic correlations for most traits, and no pairwise combination had a different sign (rg and rp). The results indicate that the CT and LAI traits had the greatest significant negative correlation (rg = −0.719 and rp =−0.704). Additionally, the results indicated that the highest significant positive correlations were between DH and DM (rg = 0.842 and rp = 0.835), between FLA and GLA (rg = −0.845 and rp = −0.738), and between E and Gs (rg = 0.902 and rp = 883). This may indicate that there is multicollinearity between these traits and/or that they could be combined into co-factors in the factor analysis.

### 3.4. Selected Genotypes Based on the MSTI

#### 3.4.1. Loadings and Factor Delineation

Six principal components were maintained (Eigenvalue > 1). These explained 88.40% of the total variation among the traits (Table 3). After the varimax rotation, the commonality ranged from 0.544 (LWC) to 0.949 (DH) with an average value of 0.844, indicating that a high percentage of the variability of each variable was explained by these factors. Hence, the dimensionality of the data could be reduced by maintaining a high interpretive strength. The 20 studied traits were compiled into six factors (FA), as follows: FA1, the traits LAI, GLA, PH, and CT; FA2, the traits DH, DM, TKW, LWC, and FLA; FA3, the antioxidant traits POD, PPO, and CAT; FA4, the traits GY and GFD; FA5, the grain yield-related traits NKS and NS; and FA6, the plant-leaf-related traits Gs, E, RWC, and Pn.

**Table 3.** Principal component analysis and factor analysis (factorial loadings obtained using the varimax rotation and resulting communalities).

| Principal Component Analysis (PCA) | | | | | | | | |
|---|---|---|---|---|---|---|---|---|
| PCA | PCA1 | PCA2 | PCA3 | PCA4 | PCA5 | PCA6 | PCA7 | PCA8 |
| Eigenvalues | 4.83 | 3.96 | 2.73 | 2.40 | 1.64 | 1.31 | 0.82 | 0.61 |
| Variance (%) | 24.10 | 19.80 | 13.60 | 12.00 | 8.21 | 6.57 | 4.09 | 3.03 |
| Cum. variance (%) | 24.10 | 43.90 | 57.60 | 69.60 | 77.80 | 84.40 | 88.40 | 91.50 |

**Table 3.** *Cont.*

| | | | | | | | | |
|---|---|---|---|---|---|---|---|---|
| **Principal Component Analysis (PCA)** | | | | | | | | |
| Factor analysis (FA) | | | | | | | | |
| VAR | FA1 | FA2 | FA3 | FA4 | FA5 | FA6 | Communality | Uniquenesses |
| LAI | **−0.904** | 0.220 | 0.042 | −0.267 | −0.061 | −0.020 | 0.943 | 0.057 |
| GLA | **−0.837** | −0.147 | −0.094 | 0.048 | 0.282 | −0.329 | 0.920 | 0.080 |
| PH | **0.778** | −0.171 | 0.243 | −0.098 | 0.083 | −0.106 | 0.722 | 0.278 |
| CT | 0.752 | −0.010 | 0.086 | 0.172 | 0.212 | 0.255 | 0.712 | 0.288 |
| DH | 0.215 | **−0.899** | 0.213 | −0.155 | 0.157 | −0.006 | 0.949 | 0.051 |
| DM | 0.235 | **−0.871** | 0.013 | 0.270 | 0.055 | 0.227 | 0.940 | 0.060 |
| TKW | −0.186 | **−0.824** | 0.039 | 0.299 | −0.094 | 0.042 | 0.815 | 0.185 |
| LWC | 0.050 | **−0.631** | −0.257 | −0.083 | −0.009 | −0.266 | 0.544 | 0.456 |
| FLA | −0.591 | **−0.608** | −0.104 | 0.041 | 0.315 | −0.336 | 0.944 | 0.056 |
| POD | −0.040 | 0.040 | **0.889** | −0.173 | −0.075 | 0.026 | 0.830 | 0.170 |
| PPO | 0.200 | −0.038 | **0.844** | 0.362 | −0.077 | −0.072 | 0.895 | 0.105 |
| CAT | 0.370 | −0.053 | **0.728** | −0.286 | −0.223 | 0.190 | 0.838 | 0.162 |
| GY | −0.202 | 0.392 | −0.089 | **−0.751** | 0.057 | 0.171 | 0.799 | 0.201 |
| GFD | 0.006 | 0.116 | −0.339 | **0.704** | −0.162 | 0.471 | 0.873 | 0.127 |
| NKS | −0.032 | −0.017 | −0.171 | −0.216 | **0.892** | 0.140 | 0.892 | 0.108 |
| NS | 0.401 | −0.161 | −0.192 | 0.423 | **0.622** | −0.332 | 0.899 | 0.101 |
| Gs | −0.107 | −0.087 | −0.024 | 0.103 | 0.110 | **−0.915** | 0.879 | 0.121 |
| E | −0.149 | 0.042 | 0.101 | 0.025 | −0.129 | **−0.911** | 0.881 | 0.119 |
| RWC | −0.105 | −0.188 | −0.315 | −0.296 | −0.275 | **−0.722** | 0.830 | 0.170 |
| Pn | −0.103 | 0.501 | −0.062 | 0.002 | 0.269 | **−0.654** | 0.766 | 0.234 |

values in bold indicate related traits.

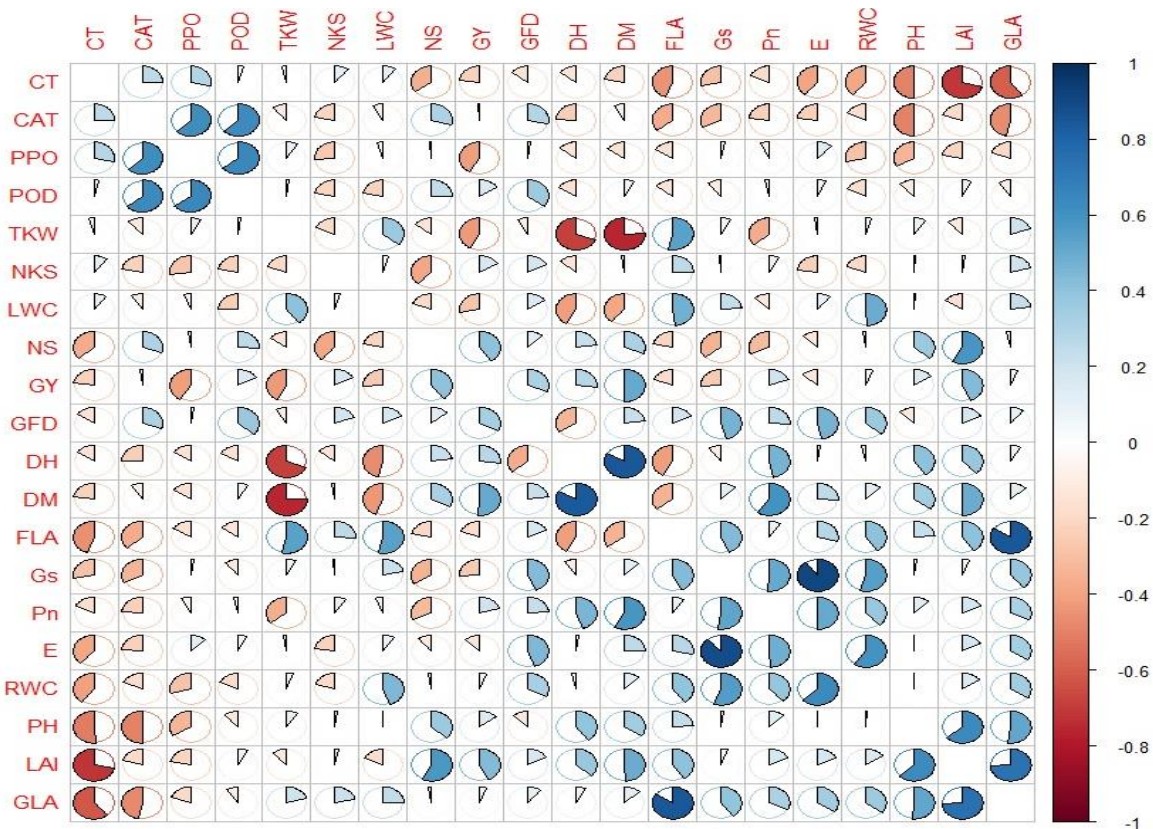

**Figure 2.** Phenotypic (upper diagonal) and genotypic (below diagonal) correlations among the 20 studied traits obtained from nine environments (n = 180). Abbreviations as described in materials and methods.

### 3.4.2. Selected Genotypes and Coincidence Index

Before any traits were removed, the genotypes selected by the MGIDI index were G17, G04, and G12; those selected by the FAI-BLUP index were G12, G13, and G06; and those selected by the Smith–Hazel index were G20, G02, and G16 (Figure 3). G13, G04, and G19 were very close to the cut-off point for index three (the red line that indicates the number of genotypes selected according to the selection pressure), which indicates that these genotypes have exciting features. Thus, further attention should be paid to the investigation of genotypes that are extremely close to the cut-off point. After the MD trait was removed for multicollinearity, the genotypes selected by the MGIDI index were G17, G04, and G11; those selected by the FAI-BLUP index were G17, G16, and G04; and those selected by the Smith–Hazel index were G06, G12, and G08. G17, G13, and G19 were extremely close to the cut-off point for three indexes. Among the genotypes selected before and after the DM trait was removed, three genotypes were selected by the MGIDI index, two genotypes were selected by the FAI-BLUP index, and one genotype was selected by the Smith–Hazel index. The genotypes G04, G13 and G17 were common to both the MGIDI and FAI-BLUP indexes before and after removal (Table 4).

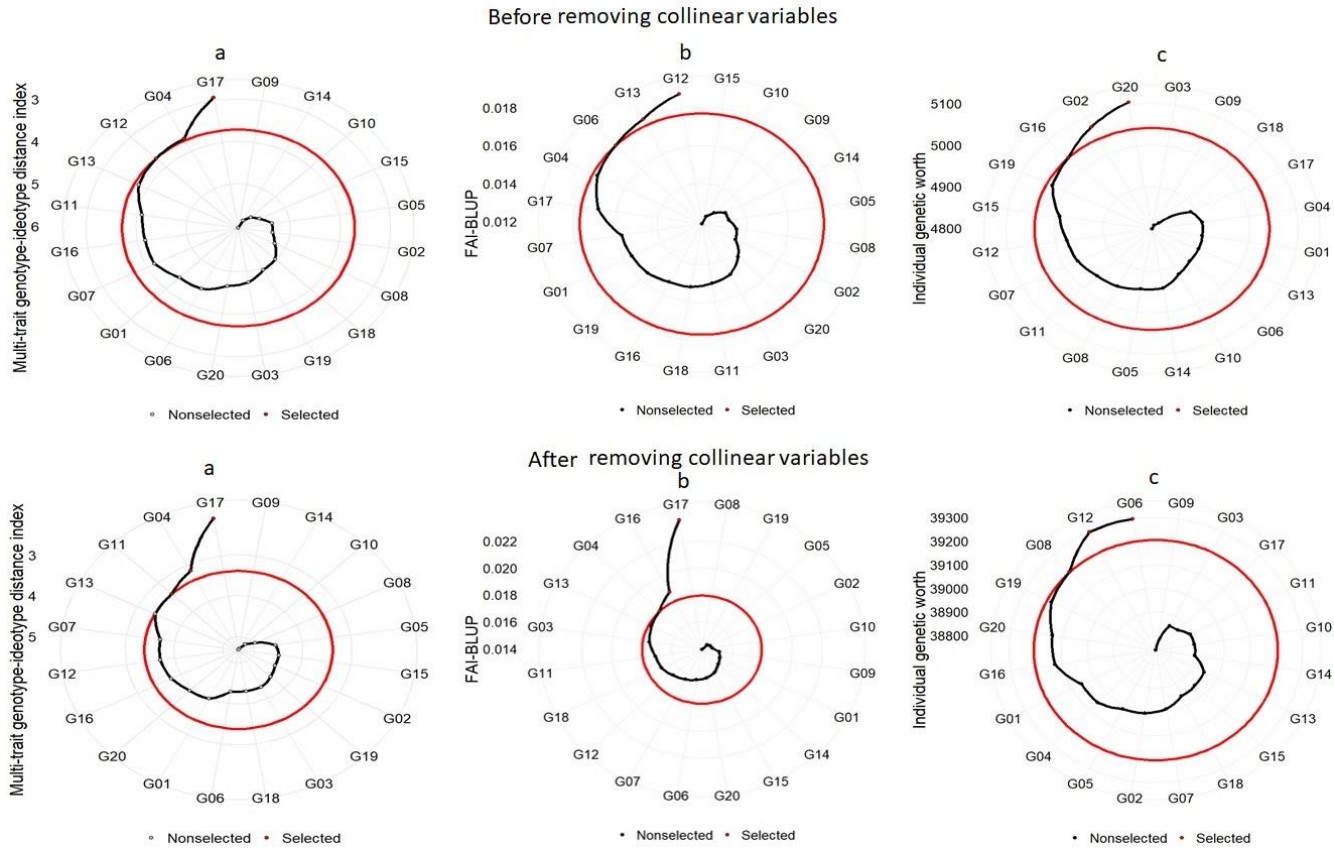

**Figure 3.** Genotype ranking for the MGIDI (**a**), FAI-BLUP (**b**), and Smith–Hazel (**c**) indices before and after removing the traits.

**Table 4.** Coincidence index (CI) of genotype selection for each pair of indices evaluated.

| Index 1 | Index 2 | CI | Genotypes |
|---|---|---|---|
| MGIDI | FAI-BLUP | 68.8 | G04,G12,G13 |
| MGIDI | Smith–Hazel | −25 | |
| MGIDI | MGIDI_collinearity | 68.8 | G17,G04,G13 |
| MGIDI | FAI-BLUP_collinearity | 68.8 | G17,G04,G13 |
| MGIDI | Smith–Hazel _collinearity | 6.25 | G12 |
| FAI-BLUP | Smith–Hazel | −25 | |
| FAI-BLUP | MGIDI_collinearity | 37.5 | G13,G04 |
| FAI-BLUP | FAI-BLUP_collinearity | 37.5 | G13,G04 |
| FAI-BLUP | Smith–Hazel _collinearity | 37.5 | G12,G06 |
| Smith–Hazel | MGIDI_collinearity | −25 | |
| Smith–Hazel | FAI-BLUP_collinearity | 6.25 | G16 |
| Smith–Hazel | Smith–Hazel _collinearity | 6.25 | G19 |
| MGIDI_collinearity | FAI-BLUP_collinearity | 68.8 | G17,G04,G13 |
| MGIDI_collinearity | Smith–Hazel _collinearity | −25 | |
| FAI-BLUP_ collinearity | Smith–Hazel _collinearity | −25 | |

### 3.4.3. Predicted Selection Gains

The selection gains (SG) between the three studied indexes before and after removal of the DM trait revealed that the trait numbers with desired gains were 15, 13, and 12 before removal and 13, 14, and 13 after removal for the MGIDI, FAI-BLUP, and Smith–Hazel indices, respectively (Table 5). These results suggest that the MGIDI was the most competent index for selecting genotypes with the desired properties. Compared to the FAI-BLUP and Smith–Hazel indices, the MGIDI gave further balanced gains for 20 (after removal) and 19 (before removal) analyzed traits, respectively. At the same time, it provided higher total gains (77.277% and 62.025% for traits that aimed to increase and −3.390% and −5.024 for traits that aimed to decrease) before and after the removal of the DM trait (Table 5).

**Table 5.** Predicted genetic gains for the MGIDI, FAI-BLUP, and Smith–Hazel indices before and after the removal of variable collinearity.

| Factor | Variable | Sense | Before Removal | | | After Removal | | |
|---|---|---|---|---|---|---|---|---|
| | | | MGIDI | FAI-BLUP | Smith–Hazel | MGIDI | FAI-BLUP | Smith–Hazel |
| FA1 | PH | decrease | 1.580 | 4.810 | −0.074 | −2.390 | −2.350 | 4.220 |
| FA1 | GLA | increase | 9.400 | 8.600 | 4.850 | 0.970 | 3.200 | 18.700 |
| FA1 | LAI | increase | 2.340 | 8.930 | 1.480 | −1.740 | −2.440 | 10.100 |
| FA1 | CT | increase | −0.078 | −0.213 | 0.016 | −0.006 | −0.007 | −0.372 |
| FA2 | DH | decrease | −2.200 | −1.650 | 3.540 | −2.140 | −2.530 | 0.101 |
| FA2 | DM | decrease | 0.055 | 0.019 | 0.569 | - | - | - |
| FA2 | FLA | increase | 13.500 | 11.400 | −1.410 | 2.380 | 9.210 | 20.400 |
| FA2 | LWC | increase | 0.078 | 0.205 | −0.217 | −0.175 | 0.020 | 0.143 |
| FA2 | TKW | increase | 1.560 | 4.070 | −6.450 | 0.585 | −0.359 | 3.740 |
| FA3 | POD | increase | 24.900 | 12.400 | −18.500 | 36.400 | 8.730 | −15.200 |
| FA3 | PPO | increase | 4.330 | −4.110 | −9.590 | 22.500 | 3.300 | −14.000 |
| FA3 | CAT | increase | −7.010 | −14.800 | −28.400 | 14.300 | −3.690 | −19.700 |
| FA4 | GFD | decrease | 1.270 | 0.908 | −0.842 | −0.494 | 1.060 | −0.323 |
| FA4 | GY | increase | 0.396 | 4.790 | 3.530 | 6.540 | 0.031 | 3.310 |
| FA5 | NS | decrease | −1.190 | 1.220 | −2.860 | 6.340 | −3.830 | 1.700 |
| FA5 | NKS | increase | 5.500 | 3.190 | 8.040 | 3.570 | 11.900 | 0.696 |
| FA6 | RWC | increase | 0.313 | 0.889 | 0.092 | −2.150 | 0.478 | 0.843 |
| FA6 | Pn | increase | 3.920 | 2.580 | 5.650 | 2.360 | 7.640 | −4.260 |
| FA6 | Gs | increase | 7.720 | 5.770 | 0.992 | 1.020 | 8.380 | 1.060 |
| FA6 | E | increase | 3.320 | 2.630 | 0.674 | −1.420 | 2.990 | 0.513 |
| Total (Increase) | | | 77.277 | 66.674 | 25.325 | 62.025 | 55.880 | 59.606 |
| Total (Decrease) | | | −3.390 | −1.650 | −3.776 | −5.024 | −8.710 | −0.222 |

### 3.4.4. Strengths and Weaknesses

Figure 4 shows the strengths and weaknesses of selected genotypes, which were calculated by the factor's contribution to the MGIDI indices of the four selected genotypes. The MGIDI was classified into two contributing factors (less and more), where the factors that contributed more were plotted near and/or inside the center, whereas factors that contributed less were plotted towards the figure's edge. FA1 had a higher contribution to the MGIDI of G17, suggesting that this genotype performs poorly for the PH, GLA, LAI, and CT traits. On the other hand, FA1 had the smallest contribution to genotypes G04, G12, and G13, indicating these genotypes were the best-performing among the selected ones for the FA1 traits. FA2 and FA4 had greater contributions to the MGIDI of G04 and smaller contributions to genotypes G12, G13, and G17, resulting in these genotypes having strengths related to FA2 and FA4. In view of the traits of FLA and TKW in FA2 and the GFD trait in FA4, positive gains are desired, so these genotypes should have (simultaneously) high values for FLA and TKW, the traits within FA2, and the GFD trait within FA4. In addition to the DH trait in FA2, these genotypes were shown to have a shorter vegetative period. The small contributions of FA3 and FA5 to G04 in the MGIDI (Figure 4) suggest that this genotype has high values for the antioxidant traits (POD, PPO, and CAT) in FA3 and the NS and NKS traits in FA5. In comparison, for G12, G13, and G17, FA3 and FA5 have high contributions. Finally, the small contribution of FA6 to G13 and G17 indicates that these genotypes have above-average values for RWC, Pn, Gs, and E.

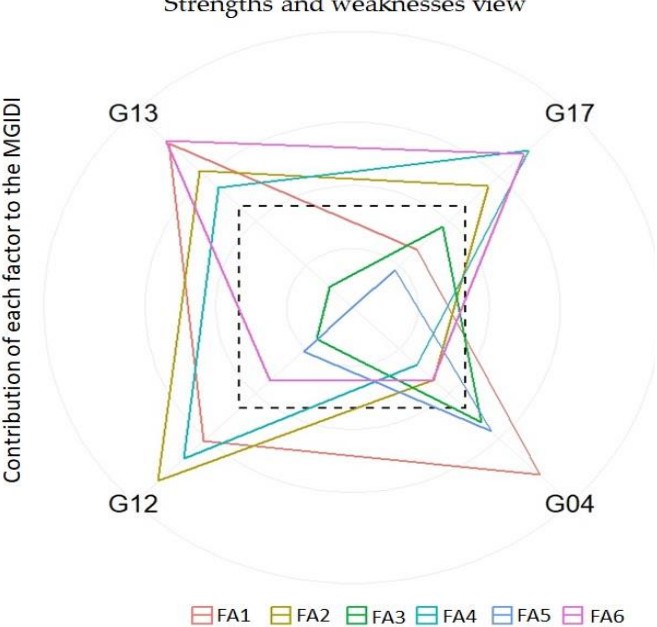

**Figure 4.** The strengths and weaknesses of the selected genotypes are shown as the proportion of each factor on the computed MGIDI.

### 3.5. Joint Regression Model of the Stability Analysis and Environmental Indices

Regarding Eberhart and Russell's model, the analysis revealed highly significant differences through a pooled analysis of variance for the main effects, genotypes, environments, and interaction effects (Table 6). The mean GY per hectare among the genotypes ranged from 3.21 (G01) to 6.39 (G04) with an overall mean of 4.53. The joint regression model of the stability analysis showed that no genotype had $b_i = 1$ and $S^2di = 0$. The genotype G18 had a bi value of near to one, showing that this genotype produced almost comparable GY values per hectare under all nine environments (Table 6). Genotypes G05 ($\mu = 4.84$, $b_i = 1.270$ **, $S^2di = 0.077$ **), G09 ($\mu = 4.86$, $b_i = 1.420$ **, $S^2di = 0.235$ **), G15 ($\mu = 4.80$, $b_i = 2.340$ **, $S^2di = 0.187$ **) and G19 ($\mu = 5.39$, $b_i = 2.150$ **, $S^2di = 0.185$ **) were observed to be stable in the control (E1, E4, and E7) environments (Table 6), whereas for genotypes G02 ($\mu = 4.63$,

$b_i$ = 0.792 **, $S^2$di = 0.209 **), G08 ($\mu$ = 4.81, bi = 0.718 **, $S^2$di = 0.292 **), G10 ($\mu$ = 5.06, $b_i$ = 0.378 **, $S^2$di = 0.133 **), G16 ($\mu$ = 4.20, $b_i$ = 0.816 **, $S^2$di = 0.140 **), and G20 ($\mu$ = 4.76, $b_i$ = 0.249 **, $S^2$di = 0.127 **), high means with lower bi values were observed. The lower bi values indicate that these genotypes show more resilience to unfavorable (drought (E2, E5, and E8) and heat (E3, E6, and E9)) environments. The RMSE of the genotypes ranged from 0.11 (G06) to 0.629 (G04), while the $R^2$ values ranged from 0.156 (G20) to 0.983 (G06). The environment index reveals the appropriateness of an environment in a precise location and can provide the basis for determining the favorable environment for the expression of the highest capacity of the genotype. The environmental indices that possessed positive values point to the favorable environments of the genotypes. As described by the environment index, E1, E3, and E7 showed the highest GY values and were found to be the most suitable and productive environments in comparison to the other environments (Table 6).

**Table 6.** Pooled analysis of variance of the 20 wheat genotypes across nine environmental for the GY (Eberhart and Russell, 1966 model).

| S.O.V | Df | Mean Sq | F Value | Pr(>F) |
|---|---|---|---|---|
| GEN | 19 | 17.6 | 38.344 | 0.0000 |
| ENV + (GEN x ENV) | 160 | 1.84 | 4.009 | 0.0000 |
| ENV (linear) | 1 | 178 | 387.800 | 0.0000 |
| GEN x ENV (linear) | 19 | 2.72 | 5.926 | 0.0000 |
| Pooled deviation | 140 | 0.459 | 10.552 | 0.0000 |
| Pooled error | 342 | 0.0435 | | |

| | | Stability parameters | | |
|---|---|---|---|---|
| GEN | GY | $b_i$ | $s^2$di | RMSE | $R^2$ |
| G01 | 3.213 | 0.710 *** | 0.013[ns] | 0.147 | 0.885 |
| G02 | 4.628 | 0.792 ** | 0.209 *** | 0.417 | 0.544 |
| G03 | 3.980 | 0.539 *** | 0.028 *** | 0.181 | 0.744 |
| G04 | 6.389 | 0.923[ns] | 0.494 *** | 0.629 | 0.416 |
| G05 | 4.842 | 1.270 *** | 0.077 *** | 0.267 | 0.882 |
| G06 | 4.434 | 1.460 *** | 0.001[ns] | 0.11 | 0.983 |
| G07 | 3.900 | 0.639 *** | 0.181 *** | 0.39 | 0.47 |
| G08 | 4.814 | 0.718 *** | 0.292 *** | 0.488 | 0.417 |
| G09 | 4.860 | 1.420 *** | 0.235 *** | 0.44 | 0.775 |
| G10 | 5.057 | 0.378 *** | 0.133 *** | 0.339 | 0.291 |
| G11 | 3.844 | 0.399 *** | 0.025 *** | 0.175 | 0.633 |
| G12 | 4.293 | 1.310 *** | 0.002[ns] | 0.115 | 0.977 |
| G13 | 4.186 | 1.320 *** | 0.126 *** | 0.33 | 0.841 |
| G14 | 3.661 | 0.902[ns] | 0.145 *** | 0.353 | 0.684 |
| G15 | 4.799 | 2.340 *** | 0.187 *** | 0.395 | 0.92 |
| G16 | 4.203 | 0.816 ** | 0.140 *** | 0.346 | 0.647 |
| G17 | 3.347 | 0.683 *** | 0.092 *** | 0.287 | 0.651 |
| G18 | 5.980 | 0.984[ns] | 0.079 *** | 0.27 | 0.814 |
| G19 | 5.393 | 2.150 *** | 0.185 *** | 0.394 | 0.908 |
| G20 | 4.764 | 0.249 *** | 0.127 *** | 0.332 | 0.156 |

| | | | | Environmental indices | | | | |
|---|---|---|---|---|---|---|---|---|
| ENV | E1 | E2 | E3 | E4 | E5 | E6 | E7 | E8 | E9 |
| Overall mean/ENV. | 5.200 | 3.980 | 4.568 | 5.374 | 4.294 | 3.760 | 5.187 | 4.037 | 4.064 |
| Index | 0.670 | −0.550 | 0.039 | 0.845 | −0.235 | −0.769 | 0.758 | −0.393 | −0.365 |
| Class | Fav [#] | Unfav [#] | fav | fav | unfav | unfav | fav | unfav | unfav |

** and *** indicate significance levels at $p < 0.01$, $p < 0.001$, respectively. ns indicates insignificance .[#] fav= favorable and unfav= unfavorable.

### 3.6. Stability Indexes Based on a Mixed-Effect Model of the Evaluated Genotypes

The HMGV describes the phenotypic stability of the GY values (Table 7). The highest values for the genotypes were obtained, in the following order, for G04, G18, G19, G10, and G20. The HMGV method relies on the prediction of the genetic values of the genotypes using the phenotypic values, taking into account the harmonic means and environmental deviation. The RPGV describes the adaptability of the genotypic values and can capitalize on the genotype response to improvements in the agricultural environment. In our study,

there was a high level of compatibility in the scores for the RPGV and HMGV, and 12 genotypes had completely identical scores. The HMRPGV describes the genetic values predicted with the BLUP method. It combines the RPGV and HMGV methods and the score for 13 genotypes were completely identical. These three methods indicated the superiority of the G04, G18, G19, and G10 genotypes, which were shown to be broadly stable and adaptable (Table 6) with identical genotype rankings with the three methods (HMVG, RPGV, and HMRPGV). The stability of the GY of the examined genotypes was assessed based on the WAASB scores. Genotype G01 (0.172) was found to be the most stable, followed by G12 (0.184), G05 (0.193), and G17 (0.230), while genotype G19 (0.679) was found to be the most unstable, followed by G15 (0.647), G18 (0.549), and G08 (0.400) for the GY.

*3.7. Understanding the Genotype × Environment Interaction*

3.7.1. Biplot Interpretation

Figure 5 provides an overview of the "which-wonwhere" pattern. The genotypes G18 and G4 won in two of the studied environments (E6 and E9) and seven of the studied environments (E1, E2, E3, E4, E5, E7 and E8), respectively. These genotypes are illustrated by a line with the formula [y= 5.98 + (−0.21 x)] for G18 and y= 6.39 + (0.45 x) for G04, where x is the (IPCA1 score) environmental. The left-most score of −1.48 refers to a yield of 6.30 t. ha−1 (for G18), while the rightmost score of 0.768 refers to a yield of 6.73 t.ha$^{-1}$(for G4). These two genotypes gave the highest yields and the smallest IPCA1 scores (−0.21 for G18 and 0.45 for G04, which defines the slope of the line) among the tested genotypes. G05 had the lowest predicted mean (4.84 t.ha$^{-1}$) but the smallest IPCA1 score (−0.0033), and its equations was (y= 4.84 + (−0.0033 x)), so it was classified it as the "universal winner" (Figure 5a).

Figure 5b demonstrates the four quadrants of the genotypes/ environment for comprehensive interpretation and a joint evaluation of the performance and stability. The first quarter included unstable genotypes that significantly contribute to the GEI in environments with a high discrimination ability. This quadrant did not include any genotypes. The second quadrant included unstable genotypes that were highly productive. The environments situated in this quadrant should be a focus, as they have high volumes of the response variables and present a high discrimination ability for the genotypes. The environments (E1, E4, and E7) were nonstressed, and the GY value was higher than the grand mean; however, the discrimination ability of the genotypes was higher in E4 (Figure 5b). The genotypes G04, G08 G15, and G19 were situated in this quadrant. Although they presented with GY values that were higher than the grand mean, they presented the highest WAASB values. Thus, private adaptations (Figure 5a) need to be investigated for these genotypes within this quadrant. The third quadrant included widely adapted and low-productive genotypes as a result of declining WAASB values, for which the decline indicates a more stable genotype performance across the environments. This included the genotypes G01, G03, G06, G07, G11, G12, G13, G14, G16, and G17. The environment E3, situated in this quadrant, can be seen as having low values for both its production and discrimination abilities. The fourth quadrant included broadly adapted genotypes with above-mean productivity values and lower WAASB values, such as the genotypes G02, G05, G09, G10, G18, and G20. In our results, only 54.45% of the GEI variance was expounded by IPCA1, but a closer look at the WAASB values indicated that G01was, in fact, more stable (smaller WAASB value), possibly due to 45.45% of the variance not being expounded by IPCA1. We previously showed that G05 had the smallest IPCA1 value, so it was more stable when using only the first IPCA.

**Table 7.** Stability indexes based on a mixed-effect model.

| Genotypes | Y | HMGV | HMGV_R | RPGV | RPGV_Y | RPGV_R | HMRPGV | HMRPGV_Y | HMRPGV_R | WAASB | WAASB_R |
|---|---|---|---|---|---|---|---|---|---|---|---|
| G01 | | 3.21 | 3.16 | 20 | 0.71 | 3.21 | 20 | 0.708 | 3.21 | 20 | 0.172 | 1 |
| G02 | | 4.63 | 4.54 | 9 | 1.03 | 4.64 | 10 | 1.02 | 4.60 | 9 | 0.321 | 10 |
| G03 | | 3.98 | 3.95 | 15 | 0.884 | 4.00 | 15 | 0.881 | 3.99 | 15 | 0.272 | 8 |
| G04 | | 6.39 | 6.26 | 1 | 1.42 | 6.42 | 1 | 1.40 | 6.34 | 1 | 0.549 | 18 |
| G05 | | 4.84 | 4.73 | 6 | 1.07 | 4.83 | 7 | 1.06 | 4.81 | 5 | 0.193 | 3 |
| G06 | | 4.43 | 4.29 | 11 | 0.972 | 4.40 | 11 | 0.968 | 4.39 | 11 | 0.257 | 6 |
| G07 | | 3.90 | 3.82 | 17 | 0.865 | 3.92 | 16 | 0.855 | 3.87 | 16 | 0.387 | 16 |
| G08 | | 4.81 | 4.73 | 7 | 1.07 | 4.84 | 5 | 1.05 | 4.78 | 7 | 0.4 | 17 |
| G09 | | 4.86 | 4.69 | 8 | 1.07 | 4.84 | 6 | 1.06 | 4.79 | 6 | 0.335 | 12 |
| G10 | | 5.06 | 5.02 | 4 | 1.13 | 5.11 | 4 | 1.12 | 5.05 | 4 | 0.339 | 13 |
| G11 | | 3.84 | 3.82 | 16 | 0.856 | 3.88 | 17 | 0.851 | 3.85 | 17 | 0.347 | 14 |
| G12 | | 4.29 | 4.17 | 12 | 0.943 | 4.27 | 12 | 0.94 | 4.26 | 12 | 0.184 | 2 |
| G13 | | 4.19 | 4.01 | 14 | 0.918 | 4.16 | 14 | 0.907 | 4.11 | 14 | 0.334 | 11 |
| G14 | | 3.66 | 3.56 | 18 | 0.807 | 3.65 | 18 | 0.8 | 3.62 | 18 | 0.26 | 7 |
| G15 | | 4.80 | 4.38 | 10 | 1.04 | 4.71 | 9 | 1.00 | 4.54 | 10 | 0.647 | 19 |
| G16 | | 4.20 | 4.12 | 13 | 0.93 | 4.21 | 13 | 0.923 | 4.18 | 13 | 0.282 | 9 |
| G17 | | 3.35 | 3.27 | 19 | 0.74 | 3.35 | 19 | 0.734 | 3.32 | 19 | 0.23 | 4 |
| G18 | | 5.98 | 5.91 | 2 | 1.32 | 6.00 | 2 | 1.32 | 5.98 | 2 | 0.242 | 5 |
| G19 | | 5.39 | 5.10 | 3 | 1.18 | 5.33 | 3 | 1.16 | 5.24 | 3 | 0.679 | 20 |
| G20 | | 4.76 | 4.73 | 5 | 1.06 | 4.82 | 8 | 1.05 | 4.76 | 8 | 0.374 | 15 |

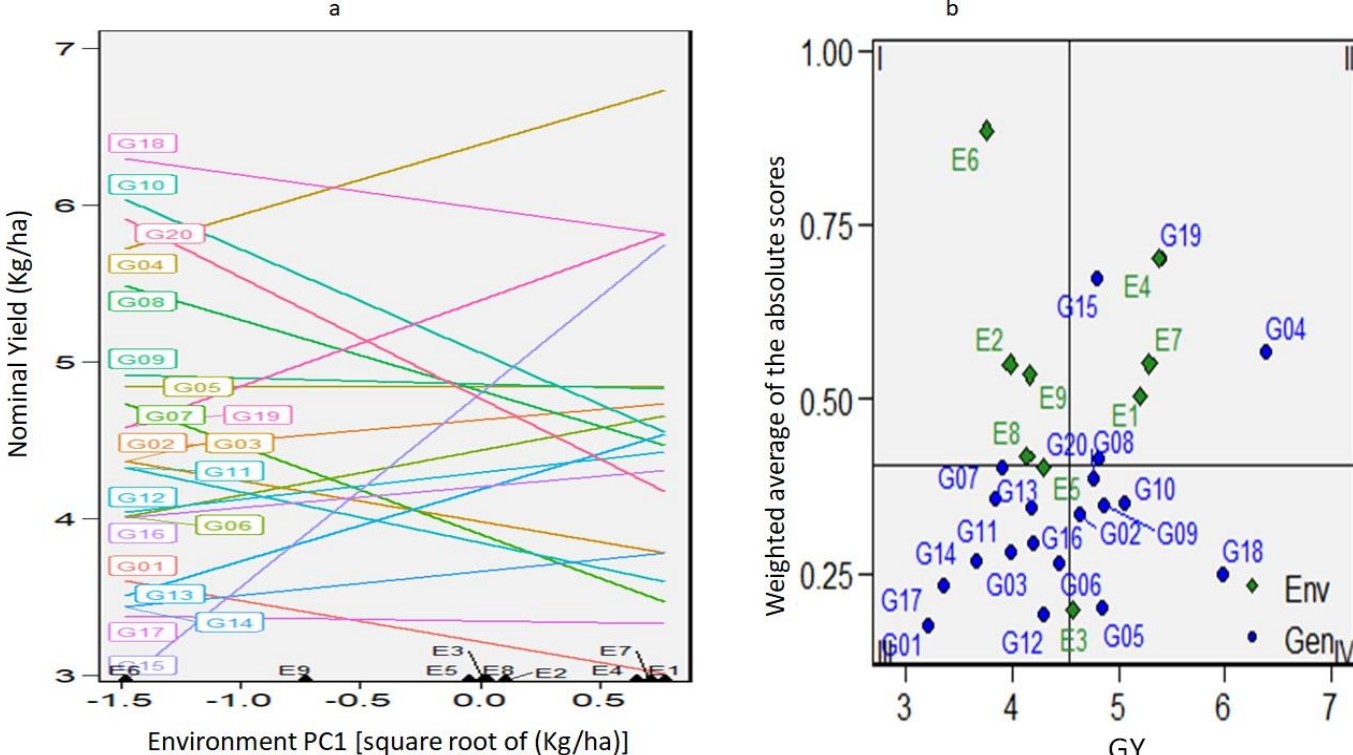

**Figure 5.** Biplot interpretation for understanding the genotype × environment interaction for the 20 wheat genotypes. (**a**) Nominal GY for the genotypes as a function of the environment scores of the IPCA1 and (**b**) biplot of the GY vs. the weighted average of absolute scores for the best linear unbiased predictions of the genotypes vs. the environment interaction (WAASB).

### 3.7.2. Genotype Ranking Depending on the Number of Retained Interaction Principal Component Axes

The ranks of the genotypes in relation to the stability depending on the number of IPCA used in the WAASB evaluation are shown in Figure 6. The eight axes were considered, and it was observed that the genotype ranking was altered depending on the extent to which IPCAs were used in the WAASB evaluation. This was sharper for four IPCAs, and genotype groups with similar performance levels and stability were identified easily by the genotype colors on the left-hand side (Figure 6). The G01, G05, G06, G03, and G02 genotypes showed the lowest WAASB values within the same cluster (considering five or more IPCAs), so they were more stable. The most visible change was in G12 when using IPCA1 in the WAASB estimation. This genotype was considered the second-most stable, but with more than five IPCAs being used, G12 was the thirtieth most stable (Figure 7). This shows the benefits of the WAASB index, which has the ability to compile the differences of all IPCAs to calculate the stability. By reference to the ASV results, we found significant convergence with the WAASBY results in terms of the ranking.

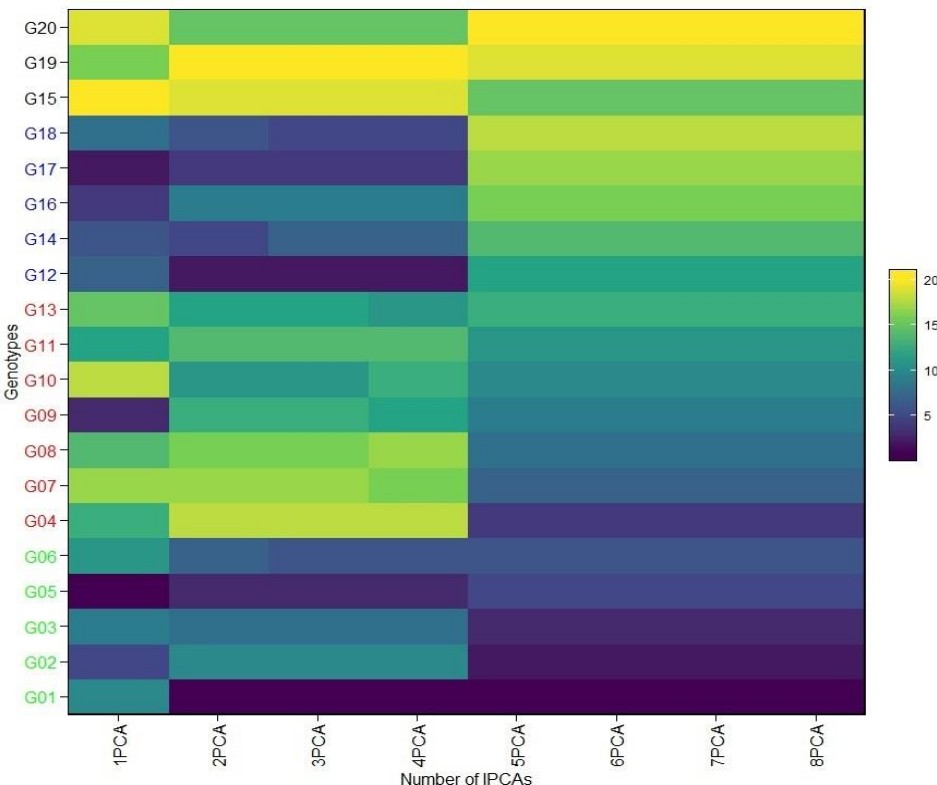

**Figure 6.** Heatmap showing the ranking of the 20 wheat genotypes in relation to the number of IPCAs used in the weighted average of the absolute scores for the BLUPs of the genotype vs. environment interaction (WAASB) estimation.

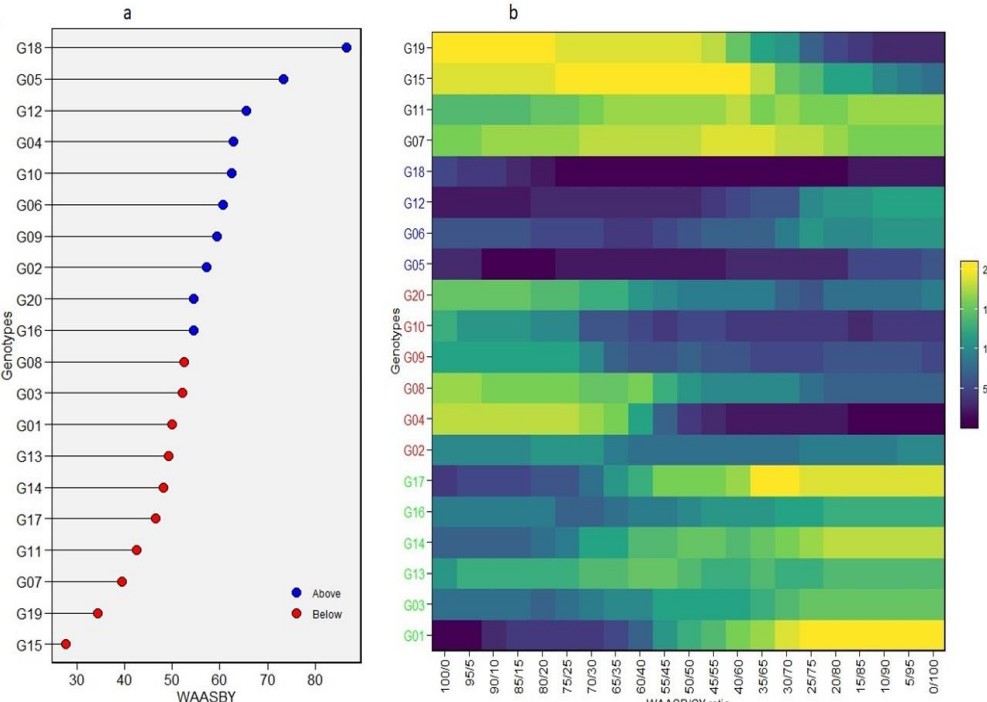

**Figure 7.** The genotype ranking depending on the weights of the stability and performance. (**a**) Estimated values of the weighted average of stability (WAASB) and mean performance (Y) (WAASBY) for the genotypes considering weights of 50 and 50 for the yield and stability, respectively. (**b**) Ranks of the genotypes considering different weights for the stability and yield.

## 4. Discussion

Abiotic stresses (drought and heat) are the major constraints to agriculture in many regions, especially in the Arab region. Wheat cultivation under drought and/or heat shows significant negative effects in terms of most morpho-physiological and agronomical traits, and this is reflected negatively in the grain yield, especially if these stresses occur during the critical periods of the plant life cycle. Therefore, the development of cultivars with high productivity coupled with high stability under varying drought and/or heat levels is the primary concern of plant breeding scientists and the long-standing goal of modern breeding programs [3,56]. In the present study, the ANOVA results demonstrated significant differences in the GE for many traits across different seasons for the three environments, and the performance of the genotypes was completely different for most traits but somewhat similar under drought and heat conditions for some other traits (Table 2). The reduction in yield under exposure to abiotic stresses compared to in the control group was very obvious for most of the genotypes studied, especially when plants were grown under heat stress (Figure 1). Many previous studies noted that wheat genotypes differ significantly in terms of the yield performance under poor stress conditions [3,57,58]. The statistical differences among the evaluated genotypes across the environments (seasons and growing conditions) provide the possibility of selecting the most appropriate and distinguished genotypes [57,59]. To understand this GEI complexity, statistical tools should be used appropriately to identify the most appropriate and stable genotypes under various abiotic stress conditions [3,56,59].

The variance components and the genetic parameters explained the variation between the studied traits (Table S4). The percentage of genotypic variance and the G × E interaction represented nearly 80% of the phenotypic manifestation for all traits (Figure S1), suggesting that genetic effects are the principal factor [45,60], except in the cases of the LWC and RWC traits. It is known that trait values are tightly linked to the genotype properties, and the environment is a limiting factor to high trait values in case of the quantitative type [3,24,45]. The $h^2$ assessment of the mean produced high values, reflecting a substantial increase in the genetic variation of the used genotypes, and these results are supported by an accuracy level of more than 0.87, and the high accuracy allows high predictability of the genetic value [24,45,60]. The $r_{ge}$ provided potential evidence regarding the genotype performance under cultivation environments that could be used to understand the nature of the interaction, in which a high value refers to a simple interaction and the opposite is also true (a low-value adversely affects the selection and ranking of the genotypes) [24,45,60]. In this study, the $r_{ge}$ showed high values (>0.580) for fourteen measured traits, indicating that the genotypic effect played a significant role in the inheritance of traits (Table S4). Correlation assessments are very important in breeding experiments, since a positive impact on selection could subsequently be concluded. Furthermore, enhanced outcomes could be obtained upon the assessment of genetic and phenotypic correlations [10,61,62], as their high values could reflect a high level of co-heritability, as observed for some of the traits. Our results revealed significant positive correlations for the GY with nine traits (Table S5 and Figure S2) ( in descending order, DM, LAI, NS, GFD, DH, Pn, NKS, POD, and PH), and the genetic correlations were similar to the phenotypic correlations, and both had the same sign [10,62].

Genotypes grown under varied environments are heterogeneous due to complex environmental interactions [3,63], so the genotype selection process, which remains stable across varied environments, is critical [3,43,46]. Breeding experts often try combine the desirable genotypes' traits in one new genotype, leading to a higher performance [10,44]. Genotype selection depending on the GY only is not preferred, because it leads to misinterpretations and inaccurate results [24,59]. Thus, multivariate techniques were used to assess the genotypic stability across varied environments and to elucidate the multicollinearity issues, which frequently occur when interacting with multiple traits [3,10,22,63]. In this regard, several multivariate analyses (PCA, FA, and selection indices (including Smith [19] and Hazel [18])) are becoming widely used to assemble the measured traits and select

the identified genotypes. In the current study, we used the PCA and FA to assemble the influential measured traits (Table 3), but no preferred genotype was determined. The FA enabled the reduction of 20 traits to only six variables (factors) that explained 84% of all studied traits. Moreover, the FA created orthogonal axes among factors, giving rise to trait scores without multicollinearity. This dimensional reduction supports breeders in the interpretation of results and in their decision making [10,44]. Olivoto and Nardino [10] enabled the development of a new index that combines all selected traits (MTSI) that favorably satisfy the plant breeder and achieve the desired goal of identifying stable genotypes with multiple traits that are suitable for more extensive adaptation by the MGIDI index.

In our study, the MGIDI index was superior to the FAI-BLUP and SH indexes in the process of selecting traits with intended gains, whether before and/or after removal of the multicollinearity trait (DM trait) as well as having greater computational efficacy [10]. The main advantage of the MGIDI compared to the SH and FAI-BLUP indexes is that the sense is identified by the breeder before starting the index computation [10]. Correlated data are abundant (high correlation) in breeding experiments, which negatively affects the selection success, especially when using the SH index, which often has a multicollinearity issue [61,64,65]. We observed (before the removal of the multicollinearity trait) that the SH with high collinearity for the DM trait offered undesirable gains for 8 of the 20 studied traits, and after removal (DM trait), only six of the 19 traits had undesirable gains (Table 3). In contrast, MGIDI index takes multicollinearity in mind, offering a high success rate in selecting traits with desirable gains, and determining the strengths and weakness of the genotypes [10]. Accordingly, when the 20 wheat genotypes were ranked based on the MSTI (Figure 3a), the genotypes G17, G04, and G12 were selected as stable genotypes under the abiotic stress conditions (drought and heat) and growing seasons (in total, nine environments). Additionally, genotype G13 was very close to the cut-off point, indicating that this genotype can display intriguing features. Strong interest in genotypes that are close to the cut-off point is crucial [10,44,63]. From a comprehensive analysis of the results, we found that four physiological traits (GLA, LAI, FLA, and Pn) and three agronomical traits (DH, TKW and NKS) combined a high level of broad-sense heritability, high-performance in the FA, and desirable gains with the MGIDI index, signifying that they deserve attention in future programs to examine to stable genotypes in the early phases. Olivoto and Nardino [10], Benakanahalli et al. [63], and Pour-Aboughadarehet al., [44] used the MGIDI index to identify ideal genotypes; it is anticipated that this index will become widely used to investigate plant crops.

The degree of the MGIDI index described by each factor is an essential tool to determine the identity of selected genotypes in terms of their strengths and weaknesses, as shown in Figure 4. It can be described as effective, straightforward, and objective [55]. From the breeders' opinion, this graph allows the selected genotypes with one trait and/or more traits requiring improvement to be identified. The MGIDI classifies the contributing factors (less and more) into factors that contribute more (plotted close to or at the center), or less (plotted towards the figure's edge). These contributions can be used to select the donors' parents in future crossbreeding programs. For example, in our future study, these genotypes G04 and G17 are expected to crossbreed with the aim of obtaining a new recombination by combining all traits into an ideotype. G04 gives the traits of DH, DM, FLA, LWC, and TKW from FA2; GFD, and GY from FA4; and RWC, Pn, Gs, and E from FA6, and G17 gives the traits of PH, GLA, LAI, and CT from FA1; POD, PPO, and CAT from FA3; and NKS and NS from FA5. If this goal is achieved, the breeder can obtain a new ideotype. The adoption of the MGIDI index in stability evaluation studies can lead to the reduction of redundant calculations and allow better strategic decisions to be made; thus, it will be easier to make recommendations for superior crop cultivars [10,63].

Because the major purpose of breeding programs is to ensure genetic stability and stabilize the yield in various environments, we tested 20 wheat genotypes across nine environments to test their adaptability in the target environments. The pooled analysis of variance for the GY stability trait, as per Eberhart and Russell [36], showed that the

variance was significant for both the main effects, genotype and environment, and the interaction effect, indicating that the performance of the genotype varied among environments. Many other researchers have reached the same result [3,44,46,66–68]. In this study, the two genotypes G06 and G12 had regression coefficients of >1.0 and were observed to be stable in favorable environments. According to the E-R model, the genotypes with a slope of >1.0 with a high mean and insignificant S2di are suitable for use in favorable environments (Table 6), and those with a slope of <1.0 are suitable for use in unfavorable environments [46,69,70]. Notably, there were no stable genotypes (bi = 1 and $S^2$di = 0.00), but the G18 genotype was almost stable. Moreover, genotype G01 was determined to be suitable for unfavorable environments [71]. The novel WAASB model explains the GEI, which combines the AMMI and BLUP models into a unique index and is used for the selection of genotypes with simultaneous high performance and stability characteristics [23,24,44,45]. The WAASB quantifies the stability of genotypes by taking into account all IPCAs from the singular value decomposition (SVD) of the matrix of GEI effects resulting from a linear mixed-effect model [24]. Our findings indicate that the minimum WAASB values were obtained for G01, G12, G05, and G17, but these values were not identical to the HMVG, RPGV, or HMRPGV scores (Table 7).

In order to achieve more robust stability and productively results, a biplot depending on the WAASB and GY values was provided (Figure 5). The principal advantage of the WAASB biplot over the AMMI biplot is that all IPCA axes are usable, hence allowing the GEI patterns not maintained in IPCA1 to be taken into account in the genotype ranking [24,44]. Additionally, the WAASB is dependent on absolute deviations. This is the opposite of the ASV, which is dependent on squared deviations, reducing the number of outliers. The WAASB gives more reliable results due to the smaller contribution of the obtained deviations to the last axes [24]. The WAASB × GY biplot (Figure 5) allows a combined explanation of the stability and productively in a two-dimensional plot, taking into account all of the IPCAs of the model (Figure 6). This may be a promising way to determine high-yielding genotypes and to adapt genotypes in future studies. In recent years, the AMMI1 biplot (as ASV) has been used most often to quantify the stability [24,72,73]. However, it was found to be unsuitable when one or even two IPCAs are used compared to the WAASB (IPCAs are used), unless the scores are very low for the genotype in the first two IPCAs. Olivoto et al. [24] showed a similar result for the G2 genotype. Previously weighting between the mean performance and stability with the WAASBY was rarely used due to the difficulty associated with weighting between the mean performance and stability (Yan and Kang, 2003). In comparison, this method now appears to be promising [24]. The WAASBY analysis was found to be a beneficial contemporaneous selection index for use in outlook studies in the MET. This is very important because, depending on the breeding program making recommendations on the genotype, the breeder may want to give preference to genotype productivity at the expense of stability or vice versa. Thus, the breeders should make use of Figure 7 when making recommendations and decisions about the selected genotype in addition to identifying the genotype group that is close to the mean performance and stability. Given the increasing demand for a greater GY due to the steady increase in the population, coupled with severe climate changes, obtaining desirable genotypes with high productivity and stability under environmental stresses is considered a perfect solution [3,44,74].

## 5. Conclusions

Information about the variation in morpho-physiological and agronomic traits is critical for determining the scale of abiotic stress tolerance in traits. Many plants showed genetic variability among the wheat genotypes used under water deficit and heat stress conditions. G04, G12, G13, and G17 were selected among the 20 genotypes as convenient and stable genotypes using the MGIDI index under all environments. By using MET trials to examine the yield performance, the WAASB index selected the genotypes G01, G05, G12, and G17 and the WAASBY index selected the genotypes G01, G05, G12, and

G17 as the superior genotypes with the greatest stability in terms of the GY. These can be recommended for cultivation under water deficit and heat stress conditions. The combination of indices (MGIDI and WAASB) and (MGIDI and WAASBY) identified the genotypes G12 and G17 and G04 and G12, respectively, as the most stable candidates. Therefore, these are considered to be novel genetic resources for improving the productivity and stabilizing the GY in wheat programs under optimal conditions and water deficit and heat stress conditions. The genotype G12 was jointly expressed in all three indices. These genotypes can be recommended as new genetic resources for improving and stabilizing the GY in wheat programs under optimal, water deficit, and heat stress conditions. Hence, these methods, if jointly used can serve as a powerful tool to assist breeders in MET.

**Supplementary Materials:** The following supporting information can be downloaded at: https://www.mdpi.com/article/10.3390/agronomy13020585/s1, Table S1. Names and pedigree of the 20 bread wheat genotypes (6 cultivars and 14 doubled haploid lines (DHLs)) used in this study; Table S2. Details of genotype codes and names of 20 bread wheat genotypes (6 cultivars and 14 doubled haploid lines (DHLs)) used in this study; Table S3. Monthly agro-climatological data at the experimental location during the growing seasons.; Table S4. Deviance analysis, estimated variance components and genetic parameters for grain yield of 20 wheat genotypes evaluated in nine environments.; Table S5. Phenotypic (upper diagonal) and genotpic (below diagonal) correlations among 20 studied traits obtained from nine environments (n = 180). Figure S1. Estimated variance components for 20 studied traits for 20 wheat genotypes in nine environments.

**Author Contributions:** Conceived and designed the experiments: I.A.-A. Performed the experiments: I.A.-A., A.I. and M.S. (Mohammed Sallam). Analyzed the data: I.A.-A. and M.S. (Mohamed Shady). Morpho-physiological measurements: I.A.-A., K.F.A., M.S. (Mohammed Sallam), I.A.-A. and M.S. (Mohamed Shady). Edited the manuscript: I.A.-A. Final approval of the version to be published: I.A.-A. and S.S.A. All authors have read and agreed to the published version of the manuscript.

**Funding:** The authors extend their appreciation to the Deputyship for Research & Innovation, Ministry of Education in Saudi Arabia for funding this research work through the project no. (IFKSURG-2 -951).

**Data Availability Statement:** All data is contained within the article or Supplementary Materials.

**Acknowledgments:** The authors extend their appreciation to the Deputyship for Research & Innovation, Ministry of Education in Saudi Arabia for funding this research work through the project no. (IFKSURG-2 -951).

**Conflicts of Interest:** The authors declare no conflict of interest.

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
