# Peer review of "Detection of High-Performance Wheat Genotypes and Genetic Stability to Determine Complex Interplay between Genotypes and Environments"

_agronomy, doi:10.3390/agronomy13020585_

Round 1

Reviewer 1 Report

1. It is suggested to add detailed description to each figure.

2.d 17 November (E4, E5, E7, and 161 E8) and 25 December (E6 and E9) in 2019/20 and 2020/21.

 Does this mean 17 November (E4, E5, E7, and 161 E8) in 2019/20 and 25 December (E6 and E9) in  2020/21.

3. It is suggested that the experimental treatment be displayed in the form of tables or pictures.

4. It is suggested to display field phenotypic photos.

5. What is the basis for defining E3, E6 and E9 as the treatment of heat stress? It should be described in the text.

6. thermalstress(E3,E6,andE9asheatstress)]was this treatment timely irrigated?

Author Response

Dear Academic Editor and Reviewer  of agronomy

Thank you very much for the valuable comments obtained from the Editor and Reviewers of agronomy Journal. Kindly, find our response to the comments point by point below. Also, the changes have been made in the MS as colorful or deleted.

# Reviewer 1

  • It is suggested to add detailed description to each figure.

Response: Thank you very much for suggesting this. We added detailed description.

  • “d 17 November (E4, E5, E7, and 161 E8) and 25 December (E6 and E9) in 2019/20 and 2020/21.

Response: Thank you very much for your comment. We added the information regarding the heat and drought treatments as shown in Table 1.

  • ” Does this mean “17 November (E4, E5, E7, and 161 E8) in 2019/20 and 25 December (E6 and E9) in 2020/21.”

Response: Thank you very much comment. Yes, we added the information regarding experimental treatments as shown in Table 1.

  • It is suggested that the experimental treatment be displayed in the form of tables or pictures.

Response: Thank you very much for the great and valuable comment and agree with you. We added the information regarding experimental treatments as shown in Table 1.

  • It is suggested to display field phenotypic photos.

Response: Thank you very much for suggested. We did not have photos for field phenotypic

  • What is the basis for defining E3, E6 and E9 as the treatment of heat stress? It should be described in the text.

Response: Thank you very much for the great and valuable comment and agree with you. We added the information regarding heat treatment as shown in Table 1. 

  • “thermalstress(E3,E6,andE9asheatstress)]”was this treatment timely irrigated?

Response: Thank you very much for your comment. We added the information regarding the heat treatment as shown in Table 1.

Reviewer 2 Report

The analysis is not consistently described and carried out. While the introduction raises an interesting point for evaluation, the Materials and Methods subsection revealed several inaccuracies and oversights. For example, it does not explain how the heat and drought treatments were carried out. Obviously, these treatments are not very important in what follows.

The ANOVA model (line 168) is more or less wrong since it does not provide an environment term (E), instead the term G is doubled. It could be a typo. However, in the following analysis (Table 1 and Fig. 1) a different ANOVA model is used. This model should also have been presented in the subsection Material and Methods.

The calculation of the heritability is not given in the subsection Material and Methods. The Smith-Hazel index was presented in the results section, but an economic weighting is essential for this index. This weight is not specified in Materials and Methods. Perhaps all properties carry equal weight. This is to be stated in Materials and Methods.

Unfortunately, there are many more such inconsistencies in the text.

Author Response

Dear Academic Editor and Reviewer of agronomy

Thank you very much for the valuable comments obtained from the Editor and Reviewers of Agronomy Journal. Kindly, find our response to the comments point by point below. Also, the changes have been made in the MS as colorful or deleted. 

# Reviewer 2

  • The analysis is not consistently described and carried out.

 Response: We greatly appreciate your critical observations as well as your contribution, and constructive and helpful comments to improve MS. We hope that we could address your comments by revisions made in the manuscript. We believe that the manuscript is substantially improved after making the suggested revisions and we have answered all your comments.

  • While the introduction raises an interesting point for evaluation, the Materials and Methods subsection revealed several inaccuracies and oversights. For example, it does not explain how the heat and drought treatments were carried out. Obviously, these treatments are not very important in what follows.

Response: Thank you very much for the great and valuable comment and agree with you. We added the information regarding the heat and drought treatments as shown in Table 1.

  • The ANOVA model (line 168) is more or less wrong since it does not provide an environment term (E), instead the term G is doubled. It could be a typo. However, in the following analysis (Table 1 and Fig. 1) a different ANOVA model is used. This model should also have been presented in the subsection Material and Methods.

Response: Thank you very much for the great and valuable comment and agree with you. It's really a typo. We corrected the model. In Table we analyzed each stress type separately as described in part 2.3.1. analysis of variance and it is the same model but the number of environments =3, and in Fig .1 we showed the overall mean for each stress type.  This model belonged linear mixed model to calculate genetic stability measurement with multiple environments

  • The calculation of the heritability is not given in the subsection Material and Methods.

Response: Thank you very much for your comment. We did not add it on the basis that it's a output from the packages used in this study were "metan" used for multi-environment trial analysis as per Olivoto and Lúcio (2020). We added it in Material and Methods

  • The Smith-Hazel index was presented in the results section, but an economic weighting is essential for this index. This weight is not specified in Materials and Methods. Perhaps all properties carry equal weight. This is to be stated in Materials and Methods.

Response: Thank you very much for your comment. We used Smith-Hazel index to comparable only with MGIDI index. Olivoto and Nardino (2021) enabled the development of a new index that combines all se-lected traits (MTSI) that favorably satisfy the plant breeder and achieve the desired goal of identifying stable genotypes with multiple traits suitable for more extensive adaptation by the MGIDI index. It superior to the Smith-Hazel index in the process of selecting traits with intended gains, whether before and/or after removal of multicollinearity trait. The main advantage of the MGIDI compared to Smith-Hazel index, the sense is identi-fied by the breeder before starting in the index computation. Such as doi: 10.1093/bioinformatics/btaa981

  • Unfortunately, there are many more such inconsistencies in the text.

Response: We believe that the manuscript is substantially improved after making the suggested revisions and we have answered all your comments. Our study was in the light(https://doi.org/10.1186/s42269-021-00576-0, doi.org/10.3390/genes13010156, doi.org/10.3390/agriculture12050602, doi.org/10.3390/agronomy11061221,  doi:10.2134/agronj2019.03.0221 and doi: 10.1093/bioinformatics/btaa981).

Reviewer 3 Report

Title: Detection of high-performance wheat genotypes and genetic stability for complex interplay between genotypes and environments.

This manuscript describes the results of different analysis approaches for effective and reliable detection of variation between twenty wheat genotypes, their interaction with environments, and genotypes' stability through various growing conditions (control, drought, and heat). Based on MGIDI and WAASB indices, as well as their combination, the most stable genotypes have been selected.

Presented experiments are innovative and functional for selecting the genotype of interest. The quality of written English is suitable for publication, but I am not qualified to decide on this issue. Methods used in the manuscript were described precisely.

Author Response

Dear Academic Editor and Reviewer of agronomy

Thank you very much for the valuable comments obtained from the Editor and Reviewers of Agronomy Journal. Kindly, find our response to the comments point by point below. Also, the changes have been made in the MS as colorful or deleted.

# Reviewer 3

  • This manuscript describes the results of different analysis approaches for effective and reliable detection of variation between twenty wheat genotypes, their interaction with environments, and genotypes' stability through various growing conditions (control, drought, and heat). Based on MGIDI and WAASB indices, as well as their combination, the most stable genotypes have been selected.

Response: We thank you for your encouraging comment.

Reviewer 4 Report

This article is generally well developped and constitutes a significant contribution in the characterization of wheat in relation to various environments. The results obtained give indications on the strategies for the genetic improvement of this plant. However, the introduction section is particularly too long and authors should indicate the paticularity of this present work. In addition,  figure 1 can be better presented as box plots instead of the heatmap used. 

Author Response

Dear Academic Editor and Reviewer of agronomy

Thank you very much for the valuable comments obtained from the Editor and Reviewers of agronomy Journal. Kindly, find our response to the comments point by point below. Also, the changes have been made in the MS as track change. 

# Reviewer 4

  • This article is generally well developped and constitutes a significant contribution in the characterization of wheat in relation to various environments. The results obtained give indications on the strategies for the genetic improvement of this plant.

Response: We greatly appreciate your critical observations as well as your contribution, and constructive and helpful comments to improve MS. We hope that we could address your comments by revisions made in the manuscript. We believe that the manuscript is substantially improved after making the suggested revisions.

  • However, the introduction section is particularly too long

Response: Thanks for your good comment. So, we deleted some sentences and words without prejudice substance.

  • In addition, figure 1 can be better presented as box plots instead of the heatmap used.

Response: Thanks for this comment. The box plots do not show the performance of genotypes under treatments, so showing the heatmap is better.

Reviewer 5 Report

>Abstract is lengthy need to more precise and clear information, which give reader a clear-cut information. 

>Line 153 as shown ___ means? Need to clear about their information. 

>Table 1 showed highly significant *** in control of seasons, genotype and its interaction. 

>Figure presentation is very poor especially for figure 1 and 2. Need to improve it because it does have alot of information without having clear identification of legends.

>Table and figures does have a lot of information which are intermingle with each other. Need to be more simple and clear.

>Why its important to have a lot of PCA for analysis and comparing the genotype relation with them, need to be more clear and super story information.

>Discussion is in good shape and with full information in the sense of output of the result format.

>Conclusion should be briefer and more comprehensive. 

Author Response

Dear Academic Editor and Reviewer of agronomy

Thank you very much for the valuable comments obtained from the Editor and Reviewers of agronomy Journal. Kindly, find our response to the comments point by point below. Also, the changes have been made in the MS as colorful or deleted.

# Reviewer 5

  • Abstract is lengthy need to more precise and clear information, which give reader a clear-cut information.

Response: Thanks for this comment. So, we deleted some words without prejudice substance.

  • Line 153 as shown ___ means? Need to clear about their information.

Response: Thank you very much for the great and valuable comment and agree with you. it was made clear and we added the information regarding experimental treatments as shown in Table 1.

  • Table 1 showed highly significant *** in control of seasons, genotype and its interaction. Response: Thanks for this comment. Some traits showed highly significant ***
  • Figure presentation is very poor especially for figure 1 and 2. Need to improve it because it does have alot of information without having clear identification of legends.

Response: Thanks for this comment. We added improvements

  • Table and figures does have a lot of information which are intermingle with each other. Need to be more simple and clear.

Response:  Thank you very much for your valuable comments. In fact, after my readings for this comment, I checked the manuscript in an attempt to shorten it, but he found that all the information was needed Because it's sequenced on each other.

  • Why its important to have a lot of PCA for analysis and comparing the genotype relation with them, need to be more clear and super story information.

Response: Thank you very much for the great and valuable comment. The multivariate analysis techniques and multidimensional methods use order to detect the samples’ genetic and geographical representation at high accuracy.  

  • Discussion is in good shape and with full information in the sense of output of the result format.

Response: Thank you very much for the great and valuable comment.

  • Conclusion should be briefer and more comprehensive.

Response: Thanks for this comment. So, we deleted some words without prejudice substance to become more comprehensive.

Round 2

Reviewer 2 Report

All of my concerns have been addressed by the authors.

Reviewer 5 Report

I like the improvement you did  in the manuscript